# Unconditional Diffusion Guidance

## Abstract

Classifier guidance is a recently introduced method to trade off mode coverage and sample fidelity in conditional diffusion models post training, in the same spirit as low temperature sampling or truncation in other types of generative models. Classifier guidance combines the score estimate of a diffusion model with the gradient of an image classifier and thereby requires training an image classifier separate from the diffusion model. It also raises the question of whether guidance can be performed without a classifier. We show that guidance can be indeed performed by a pure generative model without such a classifier: in what we call unconditional guidance, we jointly train a conditional and an unconditional diffusion model, and we combine the resulting conditional and unconditional score estimates to attain a trade-off between sample quality and diversity similar to that obtained using classifier guidance.

## 1 Introduction

Diffusion models have recently emerged as an expressive and flexible family of generative models, delivering competitive sample quality and likelihood scores on image and audio synthesis tasks (Sohl-Dickstein et al., 2015; Song & Ermon, 2019; Ho et al., 2020; Song et al., 2021b; Kingma et al., 2021; Song et al., 2021a). These models have delivered audio synthesis performance rivaling the quality of autoregressive models with substantially fewer inference steps (Chen et al., 2021; Kong et al., 2021), and they have delivered ImageNet generation results outperforming BigGAN-deep (Brock et al., 2019) and VQ-VAE-2 (Razavi et al., 2019) in terms of FID score and classification accuracy score (Ho et al., 2021; Dhariwal & Nichol, 2021).

Dhariwal & Nichol (2021) proposed *classifier guidance*, a technique to boost the sample quality of a diffusion model using an extra trained classifier. Prior to classifier guidance, it was not known how to generate "low temperature" samples from a diffusion model similar to those produced by truncated BigGAN (Brock et al., 2019) or low temperature Glow (Kingma & Dhariwal, 2018): naive attempts, such as scaling the model score vectors or decreasing the amount of Gaussian noise added during diffusion sampling, are ineffective (Dhariwal & Nichol, 2021). Classifier guidance instead mixes a diffusion model's score estimate with the input gradient of the log probability of a classifier. By varying the strength of the classifier gradient, Dhariwal & Nichol can trade off Inception score (Salimans et al., 2016) and FID score (Heusel et al., 2017) (or precision and recall) in a manner similar to varying the truncation parameter of BigGAN.

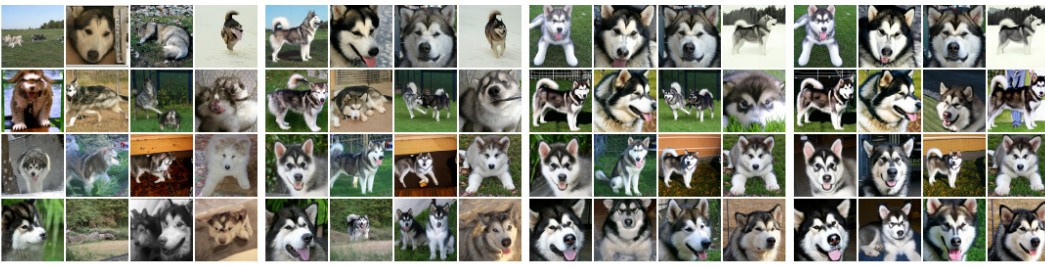

Figure 1: Unconditional guidance on the malamute class for a 64x64 ImageNet diffusion model. Left to right: increasing amounts of unconditional guidance, starting from non-guided samples on the left.

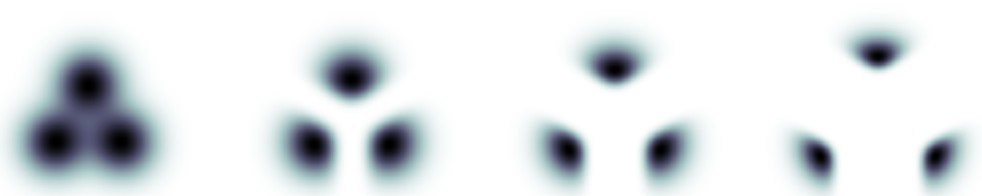

Figure 2: The effect of guidance on a mixture of three Gaussians, each mixture component representing data conditioned on a class. The leftmost plot is the non-guided marginal density. Left to right are densities of mixtures of normalized guided conditionals with increasing guidance strength.

We are interested in whether classifier guidance can be performed without a classifier. Because classifier guidance mixes a score estimate with a classifier gradient during sampling, classifier-guided diffusion sampling can be interpreted as attempting to confuse an image classifier with a gradient-based adversarial attack. This raises the question of whether classifier guidance is successful at boosting classifier-based metrics such as FID and Inception score (IS) simply because it is adversarial against such classifiers. Stepping in direction of classifier gradients also bears some resemblance to GAN training, particularly with nonparameteric generators; this also raises the question of whether classifier-guided diffusion models perform well on classifier-based metrics because they are beginning to resemble GANs, which are already known to perform well on such metrics.

To resolve these questions, we present *unconditional guidance*, our guidance method which avoids any classifier entirely. Rather than sampling in the direction of the gradient of an image classifier, unconditional guidance instead mixes the score estimates of a conditional diffusion model and a jointly trained unconditional diffusion model. By sweeping over the mixing weight, we attain a FID/IS tradeoff similar to that attained by classifier guidance. Our unconditional guidance results demonstrate that pure generative diffusion models are capable of synthesizing extremely high fidelity samples possible with other types of generative models.

## 2 BACKGROUND

We train diffusion models in continuous time (Song et al., 2021b; Chen et al., 2021; Kingma et al., 2021): letting $\mathbf{x} \sim p(\mathbf{x})$ and $\mathbf{z} = \{\mathbf{z}_\lambda \mid \lambda \in [\lambda_{\min}, \lambda_{\max}]\}$ for hyperparameters $\lambda_{\min} < \lambda_{\max} \in \mathbb{R}$, the forward process $q(\mathbf{z}|\mathbf{x})$ is the variance-preserving Markov process (Sohl-Dickstein et al., 2015):

$$q(\mathbf{z}_\lambda|\mathbf{x}) = \mathcal{N}(\alpha_\lambda \mathbf{x}, \sigma_\lambda^2 \mathbf{I}), \text{ where } \alpha_\lambda^2 = 1/(1 + e^{-\lambda}), \ \sigma_\lambda^2 = 1 - \alpha_\lambda^2 \tag{1}$$

$$q(\mathbf{z}_\lambda|\mathbf{z}_{\lambda'}) = \mathcal{N}((\alpha_\lambda/\alpha_{\lambda'})\mathbf{z}_{\lambda'}, \sigma_{\lambda|\lambda'}^2 \mathbf{I}), \text{ where } \lambda < \lambda', \ \sigma_{\lambda|\lambda'}^2 = (1 - e^{\lambda - \lambda'})\sigma_\lambda^2 \tag{2}$$

We will use the notation $p(\mathbf{z})$ (or $p(\mathbf{z}_\lambda)$) to denote the marginal of $\mathbf{z}$ (or $\mathbf{z}_\lambda$) when $\mathbf{x} \sim p(\mathbf{x})$ and $\mathbf{z} \sim q(\mathbf{z}|\mathbf{x})$. Note that $\lambda = \log \alpha_\lambda^2/\sigma_\lambda^2$, so $\lambda$ can be interpreted as the log signal-to-noise ratio of $\mathbf{z}_\lambda$, and the forward process runs in the direction of decreasing $\lambda$.

Conditioned on $\mathbf{x}$, the forward process can be described in reverse by the transitions $q(\mathbf{z}_{\lambda'}|\mathbf{z}_\lambda, \mathbf{x}) = \mathcal{N}(\tilde{\boldsymbol{\mu}}_{\lambda'|\lambda}(\mathbf{z}_\lambda, \mathbf{x}), \tilde{\sigma}_{\lambda'|\lambda}^2 \mathbf{I})$, where

$$\tilde{\boldsymbol{\mu}}_{\lambda'|\lambda}(\mathbf{z}_\lambda, \mathbf{x}) = e^{\lambda - \lambda'}(\alpha_{\lambda'}/\alpha_\lambda)\mathbf{z}_\lambda + (1 - e^{\lambda - \lambda'})\alpha_{\lambda'}\mathbf{x}, \quad \tilde{\sigma}_{\lambda'|\lambda}^2 = (1 - e^{\lambda - \lambda'})\sigma_{\lambda'}^2 \tag{3}$$

The reverse process generative model starts from $p_\theta(\mathbf{z}_{\lambda_{\min}}) = \mathcal{N}(\mathbf{0}, \mathbf{I})$. We specify the transitions:

$$p_\theta(\mathbf{z}_{\lambda'}|\mathbf{z}_\lambda) = \mathcal{N}(\tilde{\boldsymbol{\mu}}_{\lambda'|\lambda}(\mathbf{z}_\lambda, \mathbf{x}_\theta(\mathbf{z}_\lambda)), (\tilde{\sigma}_{\lambda'|\lambda}^2)^{1-v}(\sigma_{\lambda|\lambda'}^2)^v) \tag{4}$$

During sampling, we apply this transition along an increasing sequence $\lambda_{\min} = \lambda_1 < \cdots < \lambda_T = \lambda_{\max}$ for $T$ timesteps; in other words, we follow the discrete time ancestral sampler of Sohl-Dickstein et al. (2015); Ho et al. (2020). If the model $\mathbf{x}_\theta$ is correct, then as $T \to \infty$, we obtain samples from an SDE whose sample paths are distributed as $p(\mathbf{z})$ (Song et al., 2021b), and we use $p_\theta(\mathbf{z})$ to denote the continuous time model distribution. The variance is a log-space interpolation of $\tilde{\sigma}_{\lambda'|\lambda}^2$ and $\sigma_{\lambda|\lambda'}^2$ as

suggested by Nichol & Dhariwal (2021); we found it effective to use a constant hyperparameter $v$ rather than learned $\mathbf{z}_\lambda$-dependent $v$. Note that the variances simplify to $\tilde{\sigma}^2_{\lambda'|\lambda}$ as $\lambda' \to \lambda$, so $v$ has an effect only when sampling with non-infinitesimal timesteps as done in practice.

The reverse process mean comes from an estimate $\mathbf{x}_\theta(\mathbf{z}_\lambda) \approx \mathbf{x}$ plugged into $q(\mathbf{z}_{\lambda'}|\mathbf{z}_\lambda, \mathbf{x})$ (Ho et al., 2020; Kingma et al., 2021) ($\mathbf{x}_\theta$ also receives $\lambda$ as input, but we suppress this to keep our notation clean). We parameterize $\mathbf{x}_\theta$ in terms of $\boldsymbol{\epsilon}$-prediction (Ho et al., 2020): $\mathbf{x}_\theta(\mathbf{z}_\lambda) = (\mathbf{z}_\lambda - \sigma_\lambda \boldsymbol{\epsilon}_\theta(\mathbf{z}_\lambda))/\alpha_\lambda$, and we train on the objective

$$\mathbb{E}_{\boldsymbol{\epsilon},\lambda}\big[\|\boldsymbol{\epsilon}_\theta(\mathbf{z}_\lambda) - \boldsymbol{\epsilon}\|_2^2\big] \tag{5}$$

where $\boldsymbol{\epsilon} \sim \mathcal{N}(\mathbf{0}, \mathbf{I})$, $\mathbf{z}_\lambda = \alpha_\lambda \mathbf{x} + \sigma_\lambda \boldsymbol{\epsilon}$, and $\lambda$ is drawn from a distribution $p(\lambda)$ over $[\lambda_{\min}, \lambda_{\max}]$. This objective is denoising score matching (Vincent, 2011; Hyvärinen & Dayan, 2005) over multiple noise scales (Song & Ermon, 2019), and when $p(\lambda)$ is uniform, the objective is proportional to the variational lower bound on the marginal log likelihood of the latent variable model $\int p(\mathbf{x}|\mathbf{z})p_\theta(\mathbf{z})d\mathbf{z}$, ignoring the term for the unspecified decoder $p(\mathbf{x}|\mathbf{z})$ and for the prior at $\mathbf{z}_{\lambda_{\min}}$ (Kingma et al., 2021).

If $p(\lambda)$ is not uniform, the objective can be interpreted as weighted variational lower bound whose weighting can be tuned for sample quality (Ho et al., 2020; Kingma et al., 2021). We use a $p(\lambda)$ inspired by the discrete time cosine noise schedule of Nichol & Dhariwal (2021): we sample $\lambda$ via $\lambda = -2\log\tan(au + b)$ for uniformly distributed $u \in [0,1]$, where $b = \arctan(e^{-\lambda_{\max}/2})$ and $a = \arctan(e^{-\lambda_{\min}/2}) - b$. This represents a hyperbolic secant distribution modified to be supported on a bounded interval. For finite timestep generation, we use $\lambda$ values corresponding to uniformly spaced $u \in [0,1]$, and the final generated sample is $\mathbf{x}_\theta(\mathbf{z}_{\lambda_{\max}})$.

Because the loss for $\boldsymbol{\epsilon}_\theta(\mathbf{z}_\lambda)$ is denoising score matching for all $\lambda$, the score $\boldsymbol{\epsilon}_\theta(\mathbf{z}_\lambda)$ learned by our model estimates the gradient of the log-density of the distribution of our noisy data $\mathbf{z}_\lambda$, that is $\boldsymbol{\epsilon}_\theta(\mathbf{z}_\lambda) \approx \sigma_\lambda \nabla_{\mathbf{z}_\lambda} \log p(\mathbf{z}_\lambda)$. Sampling from the learned diffusion model resembles using Langevin diffusion to sample from a sequence of distributions $p(\mathbf{z}_\lambda)$ that converges to the conditional distribution $p(\mathbf{x})$ of the original data $\mathbf{x}$.

In the case of conditional generative modeling, the data $\mathbf{x}$ is drawn jointly with conditioning information $\mathbf{c}$, i.e. a class label for class-conditional image generation. The only modification to the model is that the reverse process function approximator receives $\mathbf{c}$ as input, as in $\boldsymbol{\epsilon}_\theta(\mathbf{z}_\lambda, \mathbf{c})$.

## 3 GUIDANCE

An interesting property of certain generative models, such as GANs and flow-based models, is the ability to perform truncated or low temperature sampling by decreasing the variance or range of noise inputs to the generative model at sampling time. The intended effect is to decrease the diversity of the samples while increasing the quality of each individual sample. Truncation in BigGAN (Brock et al., 2019), for example, yields a tradeoff curve between FID score and Inception score for low and high amounts of truncation, respectively. Low temperature sampling in Glow (Kingma & Dhariwal, 2018) has a similar effect.

Unfortunately, straightforward attempts of implementing truncation or low temperature sampling in diffusion models are ineffective. For example, scaling model scores or decreasing the variance of Gaussian noise in the reverse process cause the diffusion model to generate blurry, low quality samples (Dhariwal & Nichol, 2021).

### 3.1 CLASSIFIER GUIDANCE

To obtain a truncation-like effect in diffusion models, Dhariwal & Nichol (2021) introduce *classifier guidance*, where the diffusion score $\boldsymbol{\epsilon}_\theta(\mathbf{z}_\lambda, \mathbf{c}) \approx \sigma_\lambda \nabla_{\mathbf{z}_\lambda} \log p(\mathbf{z}_\lambda|\mathbf{c})$ is modified to include the gradient of the log likelihood of an auxiliary classifier model $p_\theta(\mathbf{c}|\mathbf{z}_\lambda)$ as follows:

$$\tilde{\boldsymbol{\epsilon}}_\theta(\mathbf{z}_\lambda, \mathbf{c}) = \boldsymbol{\epsilon}_\theta(\mathbf{z}_\lambda, \mathbf{c}) + w\sigma_\lambda \nabla_{\mathbf{z}_\lambda} \log p_\theta(\mathbf{c}|\mathbf{z}_\lambda) \approx \sigma_\lambda \nabla_{\mathbf{z}_\lambda}[\log p(\mathbf{z}_\lambda|\mathbf{c}) + w\log p_\theta(\mathbf{c}|\mathbf{z}_\lambda)],$$

where $w$ is a parameter that controls the strength of the classifier guidance. This modified score $\tilde{\boldsymbol{\epsilon}}_\theta(\mathbf{z}_\lambda, \mathbf{c})$ is then used in place of $\boldsymbol{\epsilon}_\theta(\mathbf{z}_\lambda, \mathbf{c})$ when sampling from the diffusion model, resulting in approximate samples from the distribution

$$\tilde{p}_\theta(\mathbf{z}_\lambda|\mathbf{c}) \propto p_\theta(\mathbf{z}_\lambda|\mathbf{c})p_\theta(\mathbf{c}|\mathbf{z}_\lambda)^w.$$

---

**Algorithm 1** Joint training a diffusion model with unconditional guidance

---

**Require:** $p_{\text{uncond}}$: probability of unconditional training
1: **repeat**
2:     $(\mathbf{x}, \mathbf{c}) \sim p(\mathbf{x}, \mathbf{c})$                              ▷ Sample data with conditioning from the dataset
3:     $\mathbf{c} \leftarrow \varnothing$ with probability $p_{\text{uncond}}$   ▷ Randomly discard conditioning to train unconditionally
4:     $\lambda \sim p(\lambda)$                                               ▷ Sample log SNR value
5:     $\boldsymbol{\epsilon} \sim \mathcal{N}(\mathbf{0}, \mathbf{I})$
6:     $\mathbf{z}_\lambda = \alpha_\lambda \mathbf{x} + \sigma_\lambda \boldsymbol{\epsilon}$                         ▷ Corrupt data to the sampled log SNR value
7:     Take gradient step on $\nabla_\theta \| \boldsymbol{\epsilon}_\theta(\mathbf{z}_\lambda, \mathbf{c}) - \boldsymbol{\epsilon} \|^2$         ▷ Optimization of denoising model
8: **until** converged

---

The effect is that of up-weighting the probability of data for which the classifier $p_\theta(\mathbf{c}|\mathbf{z}_\lambda)$ assigns high likelihood to the correct label: data that can be classified well scores high on the Inception score of perceptual quality (Salimans et al., 2016), which rewards generative models for this by design. Dhariwal & Nichol therefore find that by setting $w > 0$ they can improve the Inception score of their diffusion model, at the expense of decreased diversity in their samples.

Figure 2 illustrates the effect of numerically solved guidance $\tilde{p}_\theta(\mathbf{z}_\lambda|\mathbf{c}) \propto p_\theta(\mathbf{z}_\lambda|\mathbf{c}) p_\theta(\mathbf{c}|\mathbf{z}_\lambda)^w$ on a toy 2D example of three classes, in which the conditional distribution for each class is an isotropic Gaussian. The form of each conditional upon applying guidance is markedly non-Gaussian. As guidance strength is increased, each conditional places probability mass farther away from other classes and towards directions of high confidence given by logistic regression, and most of the mass becomes concentrated in smaller regions. This behavior can be seen as a simplistic manifestation of the Inception score boost and sample diversity decrease that occur when classifier guidance strength is increased in an ImageNet model.

Applying classifier guidance with weight $w + 1$ to an unconditional model would theoretically lead to the same result as applying classifier guidance with weight $w$ to a conditional model, because $p_\theta(\mathbf{z}_\lambda|\mathbf{c}) p_\theta(\mathbf{c}|\mathbf{z}_\lambda)^w \propto p_\theta(\mathbf{z}_\lambda) p_\theta(\mathbf{c}|\mathbf{z}_\lambda)^{w+1}$; or in terms of scores,

$$\boldsymbol{\epsilon}_\theta(\mathbf{z}_\lambda) + (w + 1)\sigma_\lambda \nabla_{\mathbf{z}_\lambda} \log p_\theta(\mathbf{c}|\mathbf{z}_\lambda) \approx \sigma_\lambda \nabla_{\mathbf{z}_\lambda} [\log p(\mathbf{z}_\lambda) + (w + 1) \log p_\theta(\mathbf{c}|\mathbf{z}_\lambda)]$$
$$= \sigma_\lambda \nabla_{\mathbf{z}_\lambda} [\log p(\mathbf{z}_\lambda|\mathbf{c}) + w \log p_\theta(\mathbf{c}|\mathbf{z}_\lambda)],$$

but interestingly, Dhariwal & Nichol obtain their best results when applying classifier guidance to an already class-conditional model, as opposed to applying guidance to an unconditional model. For this reason, we will stay in the setup of guiding an already conditional model.

### 3.2 Unconditional guidance

While classifier guidance successfully trades off IS and FID as expected from truncation or low temperature sampling, it is nonetheless reliant on gradients from an image classifier and we seek to eliminate the classifier for the reasons stated in Section 1. Here, we describe *unconditional guidance*, which achieves the same effect without such gradients. Unconditional guidance is an alternative method of modifying $\boldsymbol{\epsilon}_\theta(\mathbf{z}_\lambda, \mathbf{c})$ to have the same effect as classifier guidance, but without a classifier.

Instead of training a separate classifier model, we choose to train an unconditional denoising diffusion model $p_\theta(\mathbf{z})$ parameterized through a score estimator $\boldsymbol{\epsilon}_\theta(\mathbf{z}_\lambda)$ together with the conditional model $p_\theta(\mathbf{z}|\mathbf{c})$ parameterized through $\boldsymbol{\epsilon}_\theta(\mathbf{z}_\lambda, \mathbf{c})$. We use a single neural network to parameterize both models, where for the unconditional model we can simply input a null token $\varnothing$ for the class identifier $\mathbf{c}$ when predicting the score, i.e. $\boldsymbol{\epsilon}_\theta(\mathbf{z}_\lambda) = \boldsymbol{\epsilon}_\theta(\mathbf{z}_\lambda, \mathbf{c} = \varnothing)$. We jointly train the unconditional and conditional models simply by randomly setting $\mathbf{c}$ to the unconditional class identifier $\varnothing$ with some probability $p_{\text{uncond}}$, set as a hyperparameter. (It would certainly be possible to train separate models instead of jointly training them together, but we choose joint training because it is extremely simple to implement, does not complicate the training pipeline, and does not increase the total number of parameters.)

We can then apply Bayes' rule to obtain an *implicit classifier* as $p_\theta^i(\mathbf{c}|\mathbf{z}_\lambda) \propto p_\theta(\mathbf{z}_\lambda|\mathbf{c})/p_\theta(\mathbf{z}_\lambda)$. The score of this implicit classifier will then be given by $\nabla_{\mathbf{z}_\lambda} \log p_\theta^i(\mathbf{c}|\mathbf{z}_\lambda) \approx \frac{1}{\sigma_\lambda} [\boldsymbol{\epsilon}_\theta(\mathbf{z}_\lambda, \mathbf{c}) - \boldsymbol{\epsilon}_\theta(\mathbf{z}_\lambda)]$. Applying classifier guidance with this implicit classifier yields the following modification to the

---

**Algorithm 2** Conditional sampling with unconditional guidance

---

**Require:** $w$: guidance strength
**Require:** $\mathbf{c}$: conditioning information for conditional sampling
**Require:** $\lambda_1, \ldots, \lambda_T$: increasing log SNR sequence with $\lambda_1 = \lambda_{\min}, \lambda_T = \lambda_{\max}$
 1: $\mathbf{z}_1 \sim \mathcal{N}(\mathbf{0}, \mathbf{I})$
 2: **for** $t = 1, \ldots, T$ **do**
       $\triangleright$ Form the unconditional-guided score at log SNR $\lambda_t$
 3:    $\tilde{\boldsymbol{\epsilon}}_t = (1 + w)\boldsymbol{\epsilon}_\theta(\mathbf{z}_t, \mathbf{c}) - w\boldsymbol{\epsilon}_\theta(\mathbf{z}_t)$
       $\triangleright$ Sampling step (could be replaced by another sampler, e.g. DDIM)
 4:    $\tilde{\mathbf{x}}_t = (\mathbf{z}_t - \sigma_{\lambda_t}\tilde{\boldsymbol{\epsilon}}_t)/\alpha_{\lambda_t}$
 5:    $\mathbf{z}_{t+1} \sim \mathcal{N}(\tilde{\boldsymbol{\mu}}_{\lambda_{t+1}|\lambda_t}(\mathbf{z}_t, \tilde{\mathbf{x}}_t), (\tilde{\sigma}^2_{\lambda_{t+1}|\lambda_t})^{1-v}(\sigma^2_{\lambda_t|\lambda_{t+1}})^v)$ if $t < T$ else $\mathbf{z}_{t+1} = \tilde{\mathbf{x}}_t$
 6: **end for**
 7: **return** $\mathbf{z}_{T+1}$

---

diffusion score estimator:

$$\tilde{\boldsymbol{\epsilon}}_\theta(\mathbf{z}_\lambda, \mathbf{c}) = (1 + w)\boldsymbol{\epsilon}_\theta(\mathbf{z}_\lambda, \mathbf{c}) - w\boldsymbol{\epsilon}_\theta(\mathbf{z}_\lambda) \approx \sigma_\lambda \nabla_{\mathbf{z}_\lambda}[\log p_\theta(\mathbf{z}_\lambda|\mathbf{c}) + w \log p^i_\theta(\mathbf{c}|\mathbf{z}_\lambda)]. \qquad (6)$$

We then use $\tilde{\boldsymbol{\epsilon}}_\theta(\mathbf{z}_\lambda, \mathbf{c})$ to sample from our diffusion model as usual, thus producing approximate samples from $\tilde{p}_\theta(\mathbf{z}_\lambda|\mathbf{c}) \propto p_\theta(\mathbf{z}_\lambda|\mathbf{c})p^i_\theta(\mathbf{c}|\mathbf{z}_\lambda)^w$. Algorithms 1 and 2 describe training and sampling with unconditional guidance in detail.

Note that in some cases, it may be possible to entirely avoid training an unconditional model. If we know the class distribution and there are only a few classes, we can use the fact that $\sum_{\mathbf{c}} p(\mathbf{x}|\mathbf{c})p(\mathbf{c}) = p(\mathbf{x})$ to obtain an unconditional score from conditional scores without explicitly training for the unconditional score. Of course, this would require as many forward passes as there are possible values of $\mathbf{c}$ and would be inefficient for high dimensional conditioning signals.

It is not obvious a priori that inverting a generative model using Bayes' rule yields a good classifier that provides a useful guidance signal. For example, Grandvalet & Bengio (2004) find that discriminative models generally outperform implicit classifiers derived from generative models, even in artificial cases where the specification of those generative models exactly matches the data distribution. In cases such as ours, where we expect the model to be misspecified, classifiers derived by Bayes' rule can be inconsistent (Grünwald & Langford, 2007) and we lose all guarantees on their performance. Nevertheless, in Section 4, we show empirically that unconditional guidance is able to trade off FID and IS in the same way as classifier guidance. In Section 5 we discuss the implications of unconditional guidance in relation to classifier guidance.

## 4 EXPERIMENTS

We train diffusion models with unconditional guidance on area-downsampled class-conditional ImageNet (Russakovsky et al., 2015), the standard setting for studying tradeoffs between FID and Inception scores starting from the BigGAN paper (Brock et al., 2019).

The purpose of our experiments is to serve as a proof of concept to demonstrate that unconditional guidance is able to attain a FID/IS tradeoff similar to classifier guidance and to understand the behavior of unconditional guidance, not necessarily to push sample quality metrics to state of the art on these benchmarks. For this purpose, we use the model architectures and hyperparameters as the guided diffusion models of Dhariwal & Nichol (2021) (apart from continuous time training as specified in Section 2); those hyperparameter settings were tuned for classifier guidance and hence may be suboptimal for unconditional guidance. Furthermore, since we amortize the conditional and unconditional models into the same architecture without an extra classifier, we in fact are using less model capacity than previous work. Nevertheless, our unconditional-guided models still produce competitive sample quality metrics and sometimes outperform prior work, as can be seen in the following sections.

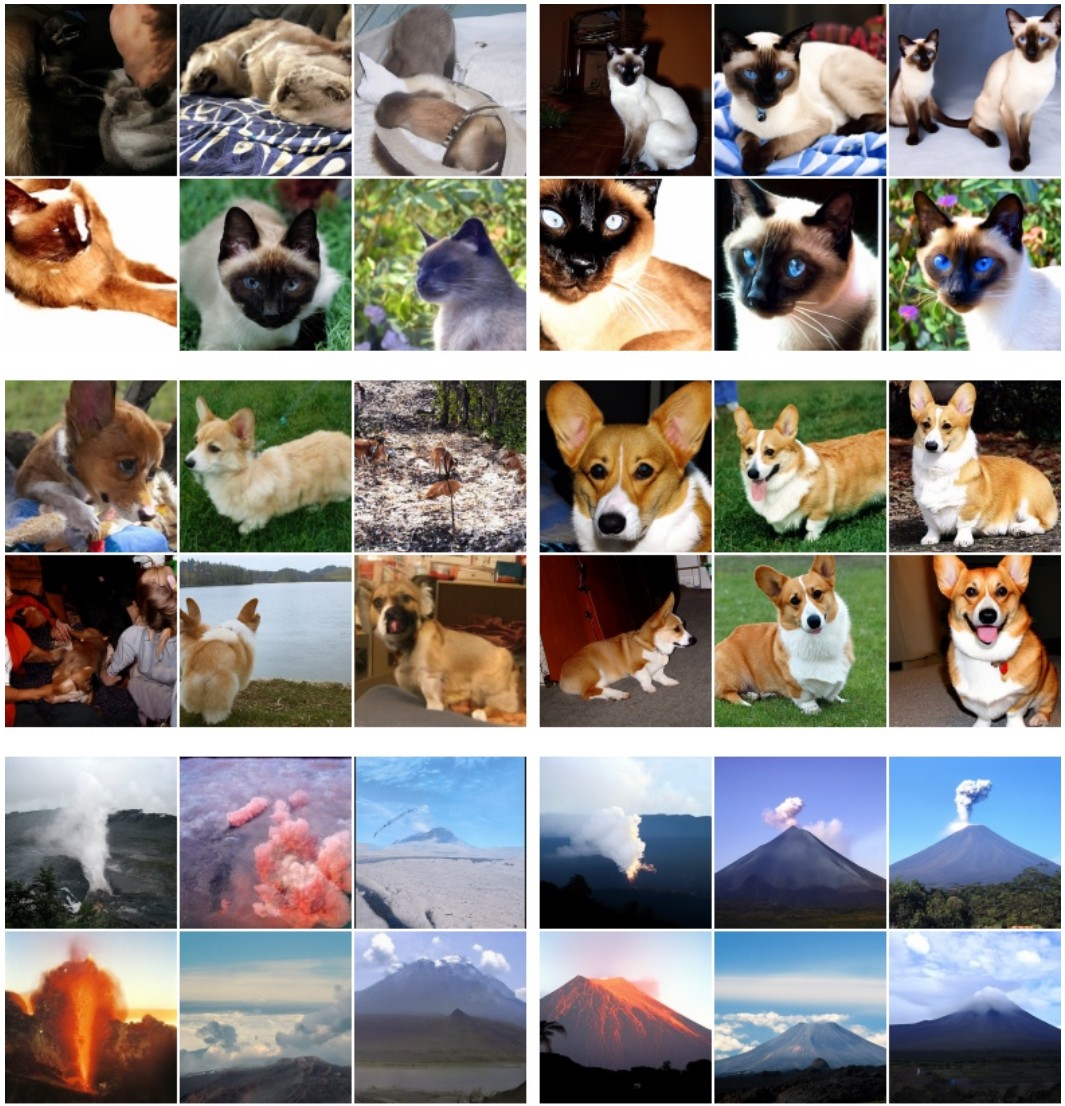

Figure 3: Unconditional guidance on 128x128 ImageNet. Left: non-guided samples, right: unconditional-guided samples with $w = 3.0$. Interestingly, strongly guided samples such as these display saturated colors. See Fig. 8 for more.

## 4.1 VARYING THE UNCONDITIONAL GUIDANCE STRENGTH

Here we experimentally verify the main claim of this paper: that unconditional guidance is able to trade off IS and FID in a manner like classifier guidance or GAN truncation. We apply our proposed unconditional guidance to $64 \times 64$ and $128 \times 128$ class-conditional ImageNet generation. In Table 1 and Fig. 4, we show sample quality effects of sweeping over the guidance strength $w$ on our $64 \times 64$ ImageNet models; Table 2 and Fig. 5 show the same for our $128 \times 128$ models. We consider $w \in \{0, 0.1, 0.2, \ldots, 4\}$ and calculate FID and Inception Scores with 50000 samples for each value following the procedures of Heusel et al. (2017) and Salimans et al. (2016). All models used log SNR endpoints $\lambda_{\min} = -20$ and $\lambda_{\max} = 20$. The $64 \times 64$ models used sampler noise interpolation coefficient $v = 0.3$ and were trained for 400 thousand steps; the $128 \times 128$ models used $v = 0.2$ and were trained for 2.7 million steps.

We obtain the best FID results with a small amount of guidance ($w = 0.1$ or $w = 0.3$, depending on the dataset) and the best IS result with strong guidance ($w \geq 4$). Between these two extremes we see a clear trade-off between these two metrics of perceptual quality, with FID monotonically decreasing

and IS monotonically increasing with $w$. Our results compare favorably to Dhariwal & Nichol (2021) and Ho et al. (2021), and in fact our $128 \times 128$ results are the state of the art in the literature. At $w = 0.3$, our model's FID score on $128 \times 128$ ImageNet outperforms the classifier-guided ADM-G, and at $w = 4.0$, our model outperforms BigGAN-deep at both FID and IS when BigGAN-deep is evaluated its best-IS truncation level.

Figures 1, 3 and 6 to 8 show randomly generated samples from our model for different levels of guidance: here we clearly see that increasing unconditional guidance strength has the expected effect of decreasing sample variety and increasing individual sample fidelity.

| Model | FID ($\downarrow$) | IS ($\uparrow$) |
|---|---|---|
| ADM (Dhariwal & Nichol, 2021) | 2.07 | - |
| CDM (Ho et al., 2021) | **1.48** | 67.95 |
| Ours | $p_{\text{uncond}} = 0.1/0.2/0.5$ | |
| $w = 0.0$ | 1.8 / 1.8 / 2.21 | 53.71 / 52.9 / 47.61 |
| $w = 0.1$ | 1.55 / 1.62 / 1.91 | 66.11 / 64.58 / 56.1 |
| $w = 0.2$ | 2.04 / 2.1 / 2.08 | 78.91 / 76.99 / 65.6 |
| $w = 0.3$ | 3.03 / 2.93 / 2.65 | 92.8 / 88.64 / 74.92 |
| $w = 0.4$ | 4.3 / 4 / 3.44 | 106.2 / 101.11 / 84.27 |
| $w = 0.5$ | 5.74 / 5.19 / 4.34 | 119.3 / 112.15 / 92.95 |
| $w = 0.6$ | 7.19 / 6.48 / 5.27 | 131.1 / 122.13 / 102 |
| $w = 0.7$ | 8.62 / 7.73 / 6.23 | 141.8 / 131.6 / 109.8 |
| $w = 0.8$ | 10.08 / 8.9 / 7.25 | 151.6 / 140.82 / 116.9 |
| $w = 0.9$ | 11.41 / 10.09 / 8.21 | 161 / 150.26 / 124.6 |
| $w = 1.0$ | 12.6 / 11.21 / 9.13 | 170.1 / 158.29 / 131.1 |
| $w = 2.0$ | 21.03 / 18.79 / 16.16 | 225.5 / 212.98 / 183 |
| $w = 3.0$ | 24.83 / 22.36 / 19.75 | 250.4 / 237.65 / 208.9 |
| $w = 4.0$ | 26.22 / 23.84 / 21.48 | **260.2** / 248.97 / 225.1 |

Table 1: ImageNet 64x64 results ($w = 0.0$ refers to non-guided models).

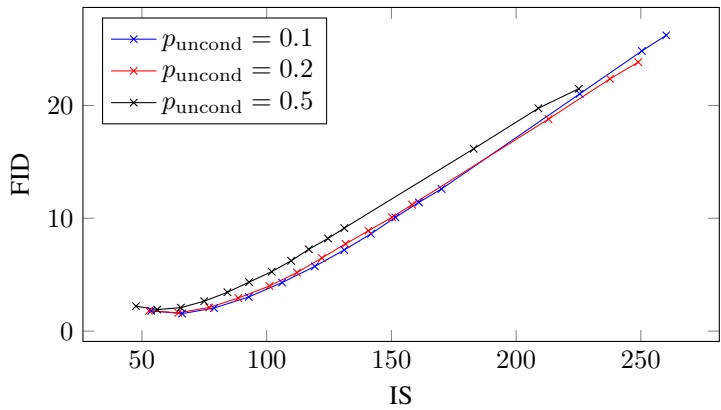

Figure 4: IS/FID curves over guidance strengths for ImageNet 64x64 models. Each curve represents a model with unconditional training probability $p_{\text{uncond}}$. Accompanies Table 1.

## 4.2 VARYING THE UNCONDITIONAL TRAINING PROBABILITY

The main hyperparameter of unconditional guidance is $p_{\text{uncond}}$, the probability of training on unconditional generation during joint training of the conditional and unconditional diffusion models. Here, we study the effect of training models on varying $p_{\text{uncond}}$ on $64 \times 64$ ImageNet.

Table 1 and Fig. 4 show the effects of $p_{\text{uncond}}$ on sample quality. We trained models with $p_{\text{uncond}} \in \{0.1, 0.2, 0.5\}$, all for 400 thousand training steps, and evaluated sample quality across various guidance strengths. We find $p_{\text{uncond}} = 0.5$ consistently performs worse than $p_{\text{uncond}} \in \{0.1, 0.2\}$ across the entire IS/FID frontier; $p_{\text{uncond}} \in \{0.1, 0.2\}$ perform about equally as well as each other.

Based on these findings, we conclude that only a relatively small portion of the model capacity of the diffusion model needs to be dedicated to the unconditional generation task in order to produce unconditional-guided scores effective for sample quality. Interestingly, for classifier guidance, Dhariwal & Nichol report that relatively small classifiers with little capacity are sufficient for effective classifier guided sampling, mirroring this phenomenon that we found with unconditionally guided models.

### 4.3 VARYING THE NUMBER OF SAMPLING STEPS

Since the number of sampling steps $T$ is known to have a major impact on the sample quality of a diffusion model, here we study the effect of varying $T$ on our $128 \times 128$ ImageNet model. Table 2 and Fig. 5 show the effect of varying $T \in \{128, 256, 1024\}$ over a range of guidance strengths. As expected, sample quality improves when $T$ is increased, and for this model $T = 256$ attains a good balance between sample quality and sampling speed.

Note that $T = 256$ is approximately the same number of sampling steps used by ADM-G (Dhariwal & Nichol, 2021), which is outperformed by our model. However, it is important to note that each sampling step for our method requires evaluating the denoising model twice, once for the conditional $\epsilon_\theta(\mathbf{z}_\lambda, \mathbf{c})$ and once for the unconditional $\epsilon_\theta(\mathbf{z}_\lambda)$. Because we used the same model architecture as ADM-G, the fair comparison in terms of sampling speed would be our $T = 128$ setting, which underperforms compared to ADM-G in terms of FID score.

| Model | FID ($\downarrow$) | IS ($\uparrow$) |
|---|---|---|
| BigGAN-deep, max IS (Brock et al., 2019) | 25 | 253 |
| BigGAN-deep (Brock et al., 2019) | 5.7 | 124.5 |
| CDM (Ho et al., 2021) | 3.52 | 128.8 |
| LOGAN (Wu et al., 2019) | 3.36 | 148.2 |
| ADM-G (Dhariwal & Nichol, 2021) | 2.97 | - |
| Ours | $T = 128/256/1024$ | |
| $w = 0.0$ | 8.11 / 7.27 / 7.22 | 81.46 / 82.45 / 81.54 |
| $w = 0.1$ | 5.31 / 4.53 / 4.5 | 105.01 / 106.12 / 104.67 |
| $w = 0.2$ | 3.7 / 3.03 / 3 | 130.79 / 132.54 / 130.09 |
| $w = 0.3$ | 3.04 / **2.43** / **2.43** | 156.09 / 158.47 / 156 |
| $w = 0.4$ | 3.02 / 2.49 / 2.48 | 183.01 / 183.41 / 180.88 |
| $w = 0.5$ | 3.43 / 2.98 / 2.96 | 206.94 / 207.98 / 204.31 |
| $w = 0.6$ | 4.09 / 3.76 / 3.73 | 227.72 / 228.83 / 226.76 |
| $w = 0.7$ | 4.96 / 4.67 / 4.69 | 247.92 / 249.25 / 247.89 |
| $w = 0.8$ | 5.93 / 5.74 / 5.71 | 265.54 / 267.99 / 265.52 |
| $w = 0.9$ | 6.89 / 6.8 / 6.81 | 280.19 / 283.41 / 281.14 |
| $w = 1.0$ | 7.88 / 7.86 / 7.8 | 295.29 / 297.98 / 294.56 |
| $w = 2.0$ | 15.9 / 15.93 / 15.75 | 378.56 / 377.37 / 373.18 |
| $w = 3.0$ | 19.77 / 19.77 / 19.56 | 409.16 / 407.44 / 405.68 |
| $w = 4.0$ | 21.55 / 21.53 / 21.45 | **422.29** / 421.03 / 419.06 |

Table 2: ImageNet 128x128 results ($w = 0.0$ refers to non-guided models).

## 5 DISCUSSION

Since unconditional guidance is able to trade off IS and FID like classifier guidance without needing an extra trained classifier, we have demonstrated that guidance can be performed with a pure generative model. We confirm that it is possible to maximize Inception scores using classifier-free unconditional guidance (and improve FID score for a small amount of guidance), thus providing evidence that classifier-based sample quality metrics can be improved using methods that are not adversarial against ImageNet classifiers using classifier gradients. Our diffusion models are parameterized by unconstrained neural networks and therefore their score estimates do not necessarily form conservative vector fields, unlike classifier gradients (Salimans & Ho, 2021). Therefore, an unconditional-guided sampler follows step directions that do not resemble classifier gradients at all and thus cannot be interpreted as a gradient-based adversarial attack on a classifier; hence our results show that boosting

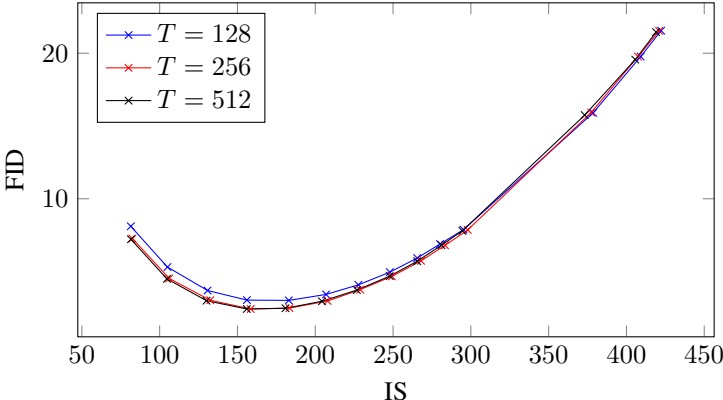

Figure 5: IS/FID curves over guidance strengths for ImageNet 128x128 models. Each curve represents sampling with a different number of timesteps $T$. Accompanies Table 2.

the classifier-based IS and FID metrics can be accomplished with pure generative models with a sampling procedure that is not adversarial against image classifiers.

We also have arrived at an intuitive explanation for how guidance works: it decreases the unconditional likelihood of the sample while increasing the conditional likelihood. Unconditional guidance accomplishes this by decreasing the unconditional likelihood with a *negative* score term, which to our knowledge has not yet been explored and may find uses in other applications.

On the practical side, unconditional guidance is extremely simple to implement and does not complicate the training pipeline of a diffusion model, unlike classifier guidance which requires training an additional classifier model. This model has to be trained on noisy data $z_\lambda$, so it is not possible to plug in a standard pre-trained classifier.

A potential disadvantage of classifier-free guidance is sampling speed. Generally, classifiers can be smaller and faster than generative models, so classifier guided sampling may be faster than classifier-free guidance because the latter needs to run two forward passes of the diffusion model, one for conditional score and another for the unconditional score. The necessity to run multiple passes of the diffusion model might be mitigated by changing the architecture to inject conditioning late in the network, but we leave this exploration for future work.

Finally, any guidance method that increases sample fidelity at the expense of diversity must face the question of whether decreased diversity is acceptable. There may be negative impacts in deployed models, since sample diversity is important to maintain in applications where certain parts of the data are underrepresented in the context of the rest of the data. It would be an interesting avenue of future work to try to boost sample quality while maintaining sample diversity.

## 6 CONCLUSION

We have presented unconditional guidance, a method to increase sample quality while decreasing sample diversity in diffusion models. Unconditional guidance can be thought of as classifier guidance without a classifier, and our results showing the effectiveness of unconditional guidance confirm that pure generative diffusion models are capable of maximizing classifier-based sample quality metrics while entirely avoiding classifier gradients. We look forward to further explorations of unconditional guidance in a wider variety of settings and data modalities.

## REPRODUCIBILITY STATEMENT

Our model architectures and training hyperparameters are from Dhariwal & Nichol (2021), which can be referenced for implementation details. Hyperparameters related to guidance are are described in Section 4. In Algorithm 1 and Algorithm 2 we provide detailed pseudocode that further describes

our implementation. We plan to open source our code and model checkpoints with the release of the final version of this paper.

## ETHICS STATEMENT

Like other machine learning methods, generative models can suffer from bias if applied to data sets that are not carefully curated. As mentioned in Section 5, any guidance or truncation method such as ours will drop sample diversity in favor of quality, and this may introduce more biases. We did not study the influence of our proposed method on this property of generative models, and careful evaluation is advised before using our algorithm in practice.

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
