# OpenReview forum: "Unconditional Diffusion Guidance"
_ICLR.cc/2022/Conference — ICLR 2022 Submitted_

### Official Review · Reviewer_4V2e · 2021-10-30

**Correctness:** 3
**Technical Novelty And Significance:** 4
**Empirical Novelty And Significance:** 2
**Recommendation:** 5
**Confidence:** 3

**Main Review:**

Even though the effectiveness of the proposed unconditional guidance is justified, I cannot admit its advantages comparing with the classifier guidance. Specifically, the proposed method need to simultaneously train an unconditional model and a conditional one, while the classifier guidance method need to simultaneously train a conditional model and a classifier. In my opinion, both of them need to train two models, so the computational cost to train is similar.  What's more, for the sampling phase, the proposed method is much slower as pointed out in Sec.5.
Therefore, I'm confused with the usefulness and the advantages of the proposed unconditional guidance.

As for the experiments, all the network architecture and hyper-parameter settings are following the previous work (Dhariwal & NIchol 2021). Through the experiments, the authors verify that the unconditional method can also achieve the goal of balancing the sample quality and diversity. However, as a complete work, it should also sufficiently exploits its best potential performance and gives some optimal settings for it, so as to provide conveniences for others to follow.

**Summary Of The Paper:**

This paper proposes an unconditional guidance, which discards the classifier in previous work.Under such unconditional guidance, it is also able to obtain a trade-off between sample quality and diversity as the classifier guidance.

**Summary Of The Review:**

In summary, I give positive comments to this work though some aspects of this method can be improved.

---

> ### Author Response · Authors · 2021-11-22
> **Response**
>
> Thank you for your constructive review.
>
> - Regarding the usefulness of our guidance method: we believe that our method’s main practical advantage is that it is extremely simple to implement our training procedure, since all that needs to be done is to randomly drop out the class label. When implemented this way, the training cost is roughly identical to training a single diffusion model. No extra training stage for an image classifier or any other model is needed.
> - Practical advantages aside, the fact that our method works answers conceptual questions about guided diffusion models. It shows that pure diffusion models, trained using denoising score matching with unrestricted neural networks (which are not necessarily the gradient of any energy function), are capable of high Inception scores, and we believe it is important to show to the community what pure diffusion models are capable of accomplishing without the assistance of anything else. Furthermore, since our method avoids any gradient-based adversarial attacks on image classifiers in the sampling process of our models, and because our denoising models have a very different structure compared to image classifier gradients, we prevent any possibility that our models are cheating these classifier-based sample quality metrics such as FID and Inception score.
> - Regarding optimal settings for hyperparameters: we will investigate this and update our paper with our findings.

---

### Official Review · Reviewer_Y1QD · 2021-11-01

**Correctness:** 3
**Technical Novelty And Significance:** 2
**Empirical Novelty And Significance:** 2
**Recommendation:** 5
**Confidence:** 3

**Main Review:**

### Positive Points

* Quite an impressive background review and explanation of Dhariwal-Nichol (2020)'s approach resembling GAN.
* Although not clearly stated in the paper, but I find replacing an explicit classifier with a generative-based implicit classifier has a potential of improving classification performance on noisy latent spaces because of better model capacity and more resemblance to the already accurate unconditional generative model.

### Negative Points

* The motivation for removing a classifier is not entirely clear. For understanding? Yes but that is not too difficult to see from the last equation/line of page 3. If the reason is to avoid adversarial attacks altogether then the reason leads to another question. Why do we need to avoid adversarial attacks? The mathematical formulation of GAN may be flawed for optimisation but the idea of increasingly harder examples to learn is beneficial. This tactic is very common in imbalanced classification problems.  More discussion on the motivation of the paper is needed.
* The paper introduced a clever trick to allow sharing weights between a conditioned model and an unconditioned model, by introducing an additional label representing the unconditional case. This works well for class labels and allows for deriving implicit classifiers directly from the model via Bayes rules. However, there is an efficiency cost at inference. For every iteration at inference, the model has to run twice, to generate conditioned and unconditioned probabilities. Using an explicit classifier, as pointed out by the authors, is faster. While I think speed is a concern, what would be more interesting is discussion, both theoretically and experimentally, of the positive point 2 above.
* The title "unconditioned diffusion guidance" is misleading. The presented model is clearly conditioned diffusion guidance because you still condition on the labels. In the end, you just replaced an explicit classifier with an implicit one. It is more appropriate to call the approach as conditioned diffusion guidance without a classifier, as mentioned in a few places in the paper.
* The explanation on page 4 that claims that a (w+1)-weight guided unconditioned model should theoretically lead to the same result as a w-weighted guided conditioned model is somewhat not entirely true. We are roughly discussing trading an unconditioned generative model and a classifier model for a conditioned generative model. If the capacity of the generative models outweighs that of a classifier model, it would be fair to say the latter would be dominated by the former.
* The experiments are somewhat lacking. Apart from the replacement of the explicit classifier with an implicit classifier sharing weights with the generative model, there is a switch from discrete-time training in Dhariwal-Nichol (2021) to continuous-time training here. The experimental results are on the total performance. However, how much of improvement of FID and IS in best-case scenarios really comes from the classifier replacement (i.e. the implicit classifier model has larger capacity than the explicit classifier, and shares weights with the generative model)? How much comes from the switch from discrete-time training to continuous-time training?
* In section 5, the argument that the proposed unconditional-guided sampler "...cannot be interpreted as a gradient-based adversarial attack on a classifier" and the explanation on how guidance works "it decreases the unconditional likelihood of the sample while increasing the conditional likelihood" somewhat contradicts with each other, since the latter can be viewed as a form of implicitly injecting adversarial attacks into the computation of the score (i.e. last equation on the 2nd-last line of page 4, section 3.2).



**Summary Of The Paper:**

The paper belongs to the class of diffusion-based generative models for generating synthetic images using class labels as guidance. It starts with a view of a recent work Dhariwal-Nichol (2021) where a classifier model is trained jointly with the diffusion-based generative model and the score (gradient of log probability) of the classifier is magnified and added to the score of the generative model to increase fidelity and the expense of diversity. In the view, Dhariwal-Nichol's approach resembles adversarial attacks of GAN-based approaches, motivating the key question that the paper addresses: whether we can train a generative model without a classifier but still enjoying the ability to trade diversity for fidelity.

The answer presented in the paper is to replace the (explicit) classifier with an implicit classifier modeled as a conditioned generative model (conditioned on the class), and the weights between the conditioned generative model than the unconditioned generative model are shared by viewing the unconditioned generative model as part of the conditional model but with an additional null class.

Experiments on ImageNet shows that the results compare favourably to Dhariwal-Nichol (2021) and Ho et al (2021), and in some cases slightly outperforming.

**Summary Of The Review:**

The paper has some novelties but they are not well-motivated and not well backed up by the provided discussion and evidences.

---

> ### Author Response · Authors · 2021-11-22
> **Response**
>
> We appreciate your detailed review.
> - “The motivation for removing a classifier is not entirely clear”: as we state in our responses to other reviewers, our ultimate motivation is to show that pure diffusion models, trained using denoising score matching with unrestricted neural networks (which are not necessarily the gradient of any energy function), are capable of high Inception scores. We believe it is important to show to the community what pure diffusion models are capable of accomplishing without the assistance of anything else. Moreover, our method is extremely simple to implement (it only requires dropping out a class label during training, and does not require implementing and training a whole separate classifier), so our method can serve as a simple default choice for practitioners when trying new tasks or datasets.
> - “Why do we need to avoid adversarial attacks?” The reason to avoid adversarial attacks on classifiers is because the standard image sample quality metrics, such as Inception score and FID score, are calculated using features from an image classifier. If we avoid adversarial attacks on image classifiers in the sampling process of our models, then we prevent any possibility that our models are cheating these sample quality metrics.
> - “Efficiency cost at inference”: we are upfront about this issue in our experiment section, and it only holds when a large unconditional model is used. We hope that the ease of implementation of our joint training procedure is a positive aspect that a practitioner can consider when choosing which guidance method to use.
> - The title "unconditioned diffusion guidance" is misleading: thank you for this feedback. We can change the title to “Classifier-free guidance of diffusion models” or another title that you may suggest.
> - “The explanation [...] that claims that a (w+1)-weight guided unconditioned model should theoretically lead to the same result as a w-weighted guided conditioned model is somewhat not entirely true”: we will clarify our point which refers to the situation in which all models are perfectly trained on infinite data.
> - “How much comes from the switch from discrete-time training to continuous-time training?” We found in initial experiments that the switch had only a negligible impact on performance, and we opted for continuous time training to simplify the implementation. Note that Dhariwhal and Nichol’s hyperparameters use 4000 discrete steps during training, which is a large value for their choice of schedule and behaves very similarly to a continuous time model.
> - “the argument that the proposed unconditional-guided sampler [...] somewhat contradicts [...]”: Here we are referring to a first order adversarial attack on an image classifier, in which the gradient of the classifier necessarily produces a conservative vector field. Our score vectors, however, do not constitute a conservative vector field, so they are a fundamentally different kind of object compared to classifier gradients. This is what we are referring to when we speak of avoiding adversarial attacks, and we will clarify this point in the paper.

---

> > ### Comment · Reviewer_Y1QD · 2021-11-23
> > **more clear, but not entirely**
> >
> > Thank you for your detailed response.
> >
> > I forgot to enumerate my points. My appology. Let's assume your response is enumerated from 1 to 7.
> >
> > Thanks for the explanations from point 3 to 6.
> >
> > 1. (Point 1) I think this comes down to differences in personal values. The ultimate motivation of the authors, is not valued highly by me (no offence intent here, I just mean the point, not the people). For me, it is clear that by powering up the score of $p_\theta(c|z_\lambda)$, which points towards high fidelity, one can make a tradeoff between FID score and Inception score. Whether you obtain the score explicitly via an external classifier or implicitly via Bayes rules is not as important.
> >
> > 2. (Point 2) To be honest, the answer does not make it easier for me. If you think the model can cheat, why not show some visual results pointing out that the metrics can be high but the images are terrible? Perhaps this might lead to a new and better metric than Inception and FID?
> >
> > 7. (Point 7) This idea is fine. And it goes well with the idea of powering up the score of $p_\theta(c|z_\lambda)$. It is just that these two ideas and the idea of adversarial attacks on image classifiers promote higher fidelity but you promote the two former and demote the latter, which is sort of inconsistent. Maybe more precise definitions and concrete examples are needed.

---

### Official Review · Reviewer_U6n2 · 2021-11-01

**Correctness:** 3
**Technical Novelty And Significance:** 2
**Empirical Novelty And Significance:** 2
**Recommendation:** 5
**Confidence:** 4

**Details Of Ethics Concerns:**

The authors have well addressed the ethics concerns that the proposed method may have raised.


**Main Review:**

Strengths:
(1) The paper is well written and easy to read. The method is clearly stated and I particularly like the way the authors posit some hypotheses on classifier guidance (i.e., being adversarial to the classifier) to motivate the proposed method and the resulting discussions.
(2) The idea of constructing an implicit classifier from the conditional and unconditional generative model to provide a guidance signal in diffusion models is new to me (although I have some concerns about its significance in the below comments).
(3) Experiments demonstrate the effectiveness of the proposed method in trading off sample diversity for sample quality.

Weaknesses:
(1) My first major concern is about the significance of the proposed method: is the unconditional guidance of more practical significance than the classifier guidance? The proposed unconditional guidance relies on a mixture of a conditional diffusion model and an unconditional diffusion model for the quality-diversity trade-off. 1) It means we have to train a conditional diffusion model from scratch to apply the proposed method. Wouldn’t it be much less flexible than training a noisy image classifier from scratch and directly applying it to the pre-trained unconditional diffusion model (in a plug-n-play manner)? Plus, the proposed method has worse sampling speed and no better sample quality and diversity trade-off compared to classifier guidance. 2) The authors mentioned the truncation trick in GANs as their motivating example, but the truncation trick is a post-hoc process that plugs in a pre-trained GAN while the proposed method has to train a new generative model (with labels!) to get the trade-off. On the contrary, the classifier guidance enjoys the same post-hoc property as the truncation trick (though it also needs labels during inference). In this sense, the classifier guidance more resembles the truncation trick than the proposed method.

(2) In experiments, to show the trade-off between sample quality and sample diversity, I would recommend adding the precision and recall metrics rather than solely focusing on FID and IS scores. Because the FID score does not only capture the sample diversity but also is affected by the sample quality, a precision-recall curve will be more convincing.

(3) In experiments, only ImageNet 64x64 and 128x128 are considered. I think adding experiments on ImageNet 256x256 and ImageNet 512x512 (as the classifier guidance paper did) will make the results in the paper more impressive.

(4) When introducing the equation $\epsilon(z_\lambda, c) = \sigma_\lambda \nabla_{z_\lambda}\log p(z_\lambda|c)$, is the $\sigma_\lambda$ assumed to be negative? It seems that a negative sign is missing before $\sigma$.

(5) The method name "unconditional guidance" is inappropriate because it requires training a conditional diffusion model with labels.


**Summary Of The Paper:**

This work proposed a method to trade-off sample diversity for sample quality in diffusion models, which is termed unconditional guidance. Different from the prior work called classifier guidance (Dhariwal & Nichol 2021) that relies on a classifier for providing the guidance signal, the proposed unconditional guidance mixes the score estimates of a conditional diffusion model and an unconditional diffusion model for a trade-off between sample quality and sample diversity. Experiments on ImageNet 64x64 and 128x128 showed that the proposed model can achieve the claimed quality-diversity tradeoff regarding FID and IS scores.


**Summary Of The Review:**

Although the idea is new in the context of diffusion models and the experiments support the major claim to some extent, I think 1) the proposed method is of less practical significance and less flexibility, compared with the prior work on classifier guidance, 2) the experiments can be improved to better support the claim and to make the results more impressive. Thus, my initial recommendation is not accepting the paper.

---

> ### Author Response · Authors · 2021-11-22
> **Response**
>
> Thank you for your thoughtful review.
> - Regarding the practical significance of our guidance method compared to previous work: our method is extremely simple to implement, only requiring a random dropout of the class label during training, and does not require training a whole extra image classifier. Training is therefore roughly as expensive as training a single diffusion model.
> - Other significant points of our guidance method are described in our response to reviewer zw1c; briefly, our method demonstrates that pure classifier-free diffusion models are capable of high inception scores using a non-conservative vector field for a guidance direction which is fundamentally different from the gradient of a classifier. More specifically:
>   - We have shown that pure diffusion models, trained using denoising score matching with unrestricted neural networks (which are not necessarily the gradient of any energy function), are capable of high Inception scores.
>   - Our guidance direction is able to boost sample quality while being fundamentally different from a classifier gradient (our guidance follows a non-conservative vector field, whereas classifier guidance follows a conservative vector field), thus putting to rest any doubts that classifier guidance is able to boost classifier-based sample quality scores only due to compatibility in the classifier architectures.
>   - Our unconditional guidance method is extremely simple to implement and can serve as a default choice when trying new tasks or datasets.
> - Regarding having to train a separate diffusion model from scratch: note again that the conditional and unconditional models are jointly trained by simply dropping out the class label during training, and the total training cost is that of training a single diffusion model. We believe that this is much simpler and more flexible than having to train a separate image classifier.
> - Regarding our “proposed method has worse sampling speed and no better sample quality” - sampling speed is slow only if a large unconditional model is used. Regardless, sampling speed does not affect our main point which is to demonstrate that pure diffusion models without classifiers are capable of high Inception scores.
> - Regarding classifier guidance resembling GAN truncation as a post-hoc method moreso  than our unconditional guidance method: we disagree because classifier guidance is actually performed in Dhariwal and Nichol’s paper using a *conditional* diffusion model, so class labels are indeed needed during training.
> - Precision-recall metrics: we will calculate these metrics in a final version of our paper.
> - Finally, we would like to emphasize that the results we did obtain are state-of-the-art or nearly so, and we hope that this can be seen as evidence of the significance of our contributions. Moreover, our method is extremely simple to implement (a one-line change during training and sampling), so we believe our method will be highly practical and useful to the community.
> - We will update the paper with your other suggested fixes; thank you for providing them.

---

> > ### Comment · Reviewer_U6n2 · 2021-11-24
> > **Thanks for the response!**
> >
> > Thanks the authors for responding to my concerns. I appreciate that the authors highlight the simplicity of the proposed method and the fact that there is little extra training cost, which I fully agree.
> >
> > - When I mentioned “train a conditional diffusion model from scratch”, I didn’t imply it is trained *separately*. Instead, I meant that applying the proposed method incurs to change the training process by incorporating the labels. This does not enjoy the plug-n-play property of both classifier guidance and truncation trick in GANs. Similar to classifier guidance, the proposed method also need extra labels for the quality-diversity. Thus, I still don’t see much of its practical significance, compared with classifier guidance.
> >
> > - Regarding the claim “classifier guidance is actually performed in Dhariwal and Nichol’s paper using a conditional diffusion model”, if I understand correctly, classifier guidance in Dhariwal and Nichol’s paper can be applied to both unconditional and conditional diffusion models. It’s just about the fact that using conditional ones can get better results.
> >
> > Therefore, I still have the concern on the significance of the proposed method, as compared with classifier guidance. After reading the authors’ response, I appreciate the simplicity of the proposed method and the discussed insights, and thus increase my score to 5.

---

### Official Review · Reviewer_zw1c · 2021-11-02

**Correctness:** 3
**Technical Novelty And Significance:** 2
**Empirical Novelty And Significance:** 3
**Recommendation:** 6
**Confidence:** 4

**Main Review:**

Strengths:

-The paper is clear and overall well-written. The experiments are easy to read
 and illustrate the announced trade-off between FID and IS.

-The idea of using implicit classifiers is interesting and provides a new step
 toward defining guided score-based generative models using only score-based
 models. Similarly, I found the intuitive explanation for how guidance works to
 be well motivated and illustrated with the implicit classifier guidance.

-The experiments are interesting and clearly illustrate the trade-off between IS
 and FID.

Weaknesses:

-In my opinion, the contribution is quite incremental in the sense that [2]
 already used classifiers for the guidance of score-based models. The authors
 justify the use of unconditional classifiers by claiming that the use of the
 classifiers of [2] can be interpreted as an adversarial attack and that
 therefore the (guided) score-based generative modeling behaves like GANs. I am
 not fully convinced by this explanation as I think that even though the model
 in [2] incorporate a guidance classifier term there are still quite far from
 being close to GANs. Also, I am not clear as of why the classifier used in [2]
 would constitute an adversarial attack against the model. As of now, the
 motivation for considering implicit classifier as presented in the introduction
 seems a bit superficial. I think the paper would truly benefit an investigation
 of the shortcomings of the score-based generative models using classifier
 guidance. Then, we could fully appreciate the benefits of using unconditional
 guidance diffusion models.

-Even though the paper is mostly experimental I think that the theoretical part
 of the paper could have been more developed. In particular it is not clear to
 me what are the properties the model given in the equation at the end of
 p.3. Also, I think that in order to be able to say that the classifier guidance
 model appoximately sample from $\tilde{p}_\theta(z_\lambda|c)$ one needs to
 explicitly write down what is the joint forward model and how to derive the
 backward model. The approximation can then be obtained similarly to Equation 4
 in [3].

General comments:

-In Table 1, I would have expected non-guided model to have better FID than the
 guided model with w=0.1. However, this does not seem to be the case. Similarly,
 in Table 2 while the IS score keeps increasing with the parameter w it is not
 the case for FID. Is there a reason why the best FID score is reached for w=0.3?

-I think it would have been interested to show the nearest neighbors in the
 ImageNet dataset depending on the parameter w. How close is the model to the
 original dataset for large values of w?

-For large values of w the score given by Equation (6) is approximately scaled
 by w. Thinking in terms of diffusions this amounts to scale the drift by a
 parameter w. The Lipschitz constant of the drift is also multiplied by w and
 therefore in order to obtain a stable discretization we need to divide the
 stepsize in the Euler-Maruyama discretization by a parameter w. However in the
 experiments it seems that the authors choose the same stepsize for every value
 of w. Could you comment on this?

[1] Song, Sohl-Dickstein, Kingma, Kumar, Ermon, Poole -- Score-based Generative Modeling through Stochastic Differential Equations

[2] Dhariwal, Nichol -- Diffusion Models Beat GANs on Image Synthesis

[3] De Bortoli, Thornton, Heng, Doucet -- Diffusion Schrodinger Bridge with Applications to Score-Based Generative Modeling

**Summary Of The Paper:**

In this paper the authors propose an improvement for score-matching based
generative modeling [1] resembling low temperature sampling as in GANs or
flow-based models. Similarly to [2] they propose to modify the drift function
used in the sampling step of the diffusion model by including the gradient of
some classifier. However, contrary to [2] the classifier considered by the
authors is implicit in the sense that it is purely defined by a conditional and
an unconditional generative model. The authors show that using such a classifier
allows to control a trade-off between IS and FID on the ImageNet dataset. The
use of such implicit classifier also provides intuition on the guidance
influence in score-based generative modeling: the model tries to reduce the
unconditional likelihood while increasing the conditional likelihood.

[1] Song, Sohl-Dickstein, Kingma, Kumar, Ermon, Poole -- Score-based Generative Modeling through Stochastic Differential Equations
[2] Dhariwal, Nichol -- Diffusion Models Beat GANs on Image Synthesis

**Summary Of The Review:**

The main idea of the paper is interesting and the experiments are quite
convincing.  However, I feel that the motivation behind this improvement is
quite superficial. I think that the authors should better motivate their study,
especially putting an emphasis on the limitations of classifier guidance in
score-based generative models. I will increase my score if the authors address
my concerns.

---

> ### Author Response · Authors · 2021-11-22
> **Response**
>
> Thank you for your constructive and detailed review.
>
> - Regarding whether our contribution is “incremental in the sense that [2] already used classifiers for the guidance of score-based models”: It certainly is true that guidance was previously proposed, and our intention is precisely to study it further. We hope that presenting an alternative yet effective form of guidance can be considered a valuable contribution. Please see the following bolded responses for a clearer explanation of why we are studying guidance.
> - “The authors justify the use of unconditional classifiers by claiming that the use of the classifiers of [2] can be interpreted as an adversarial attack and that therefore the (guided) score-based generative modeling behaves like GANs … not clear as of why the classifier used in [2] would constitute an adversarial attack against the model.”
>   - Examining the guided sampling procedure, one sees that the sample is being adjusted in the gradient direction to maximize the log probability of an image classifier. This is exactly how GAN training works, in which each sample from the generator is adjusted in the gradient direction of a discriminator, which is also an image classifier.
>   - Furthermore, increasing the log probability of an image classifier by following the  gradient direction for one class label is exactly how first-order adversarial attacks are performed. Since the image classifier has a similar architecture compared to the image classifier that is used to calculate the FID and IS metrics, there is a possibility that classifier guided sampling is hacking these metrics.
> - “The motivation for considering implicit classifier as presented in the introduction seems a bit superficial”: **our ultimate motivation is to show that pure diffusion models, trained using denoising score matching with unrestricted neural networks (which are not necessarily the gradient of any energy function), are capable of high Inception scores**. Because of our conviction that classifier-guided diffusion sampling resembles GAN training, we believe that it is important to show that classifier-free methods are capable of attaining high Inception scores just like classifier guidance in order to showcase the capabilities of pure diffusion models. Finally, note that **our guidance method is extremely simple to implement and cheap to run**: it only requires randomly dropping out the class label during training (see our training algorithm), which is much simpler to implement compared to a separate training procedure and architecture design for an image classifier.
> - We would like to emphasize is that while our guidance direction is inspired by the score of an implicit classifier, **our guidance direction is not necessarily the gradient of any classifier**. Our score estimates are derived by subtracting an unconditional score estimate from a conditional score estimate, both of which come from unrestricted neural network denoising models (which are not defined as the gradient of a scalar energy, see https://openreview.net/forum?id=9AS-TF2jRNb). Thus our guidance directions do not constitute a conservative vector field, unlike the gradient of an image classifier. Thus **one of our main contributions is showing that our guidance direction is able to boost sample quality while being fundamentally different from a classifier gradient**.
> - The remaining points also hold for classifier guidance, not just our paper, but we briefly address them here:
>   - “write down what is the joint forward model and how to derive the backward model”: indeed it is not entirely clear what the joint forward model is for classifier guidance. The best current definition for guidance is that it is simply a modified sampling scheme for a pretrained score model.
>   - “Why is FID optimal for w > 0?”: the same can be observed in the paper of Dhariwal and Nichol. We believe this occurs because FID measures a mixture of sample quality and diversity, so FID improves with a small amount of guidance because of the improved sample quality.
>   - Regarding stepsize: first, we are not strictly using a Euler-Maruyama discretization of the SDE because of our settings of the reverse transition variances (see the background section). Second, we tuned the sampler hyperparameters and found no significant differences in sample quality and therefore opted for one universal choice for simplicity.

---

> > ### Comment · Reviewer_zw1c · 2021-11-28
> > **Response to rebuttal**
> >
> > I have read the rebuttal and the other reviews.
> > I thank the authors for their clarifications.
> >
> > The authors have addressed my main concerns and I will raise my score to 6.
> > I am still a bit puzzled by the lack of theoretical results behind this procedure.
> > In particular, I'm still wondering if the scheme proposed by the authors could be seen as something else than just a modified backward sampling procedure.
> >
> > I also have to say that I strongly agree with Point 2 raised by Reviewer Y1QD and his subsequent answer: "To be honest, the answer does not make it easier for me. If you think the model can cheat, why not show some visual results pointing out that the metrics can be high but the images are terrible? Perhaps this might lead to a new and better metric than Inception and FID?"
> > I hope that the authors will address these concerns in a revised version of the paper.

---

### Decision · Program_Chairs · 2022-01-20

**Decision:**

Reject

**Comment:**

This paper modifies the conditional diffusion model guided by a classifier, as introduced by Dhariwal & Nichol 2021, by replacing the explicit classifier with an implicit classifier. This implicit classifier is derived under Bayes' rule and combined with the conditional diffusion model. This combination can be realized by mixing the score estimates of a conditional diffusion model and an unconditional diffusion model. A trade-off between sample quality and diversity, in terms of the IS and FID scores, can be achieved by adjusting the mixing weight. The paper is clearly written and easy to follow. However, the reviewers do not consider the modification to be that significant in practice, as it still requires label guidance and also increases the computational complexity. From the AC's perspective, the practical significance could be enhanced if the authors can generalize their technique beyond assisting conditional diffusion models.